# Effects of Therapeutic Hypothermia on Macrophages in Mouse Cochlea Explants

**DOI:** 10.3390/ijms24108850

**Published:** 2023-05-16

**Authors:** Werner Bader, Claudia Steinacher, Hannes Thomas Fischer, Rudolf Glueckert, Joachim Schmutzhard, Anneliese Schrott-Fischer

**Affiliations:** Department of Otorhinolaryngology, Medical University Innsbruck, 6020 Innsbruck, Austria; werner.bader@student.uibk.ac.at (W.B.); claudia.steinacher@i-med.ac.at (C.S.); hannes.fischer@tirol-kliniken.at (H.T.F.); rudolf.glueckert@i-med.ac.at (R.G.); annelies.schrott@i-med.ac.at (A.S.-F.)

**Keywords:** cochlea implant, IBA1, F4/80, CD163, CD45, immune response, inner ear, hypothermia

## Abstract

Globally, over the next few decades, more than 2.5 billion people will suffer from hearing impairment, including profound hearing loss, and millions could potentially benefit from a cochlea implant. To date, several studies have focused on tissue trauma caused by cochlea implantation. The direct immune reaction in the inner ear after an implantation has not been well studied. Recently, therapeutic hypothermia has been found to positively influence the inflammatory reaction caused by electrode insertion trauma. The present study aimed to evaluate the hypothermic effect on the structure, numbers, function and reactivity of macrophages and microglial cells. Therefore, the distribution and activated forms of macrophages in the cochlea were evaluated in an electrode insertion trauma cochlea culture model in normothermic and mild hypothermic conditions. In 10-day-old mouse cochleae, artificial electrode insertion trauma was inflicted, and then they were cultured for 24 h at 37 °C and 32 °C. The influence of mild hypothermia on macrophages was evaluated using immunostaining of cryosections using antibodies against IBA1, F4/80, CD45 and CD163. A clear influence of mild hypothermia on the distribution of activated and non-activated forms of macrophages and monocytes in the inner ear was observed. Furthermore, these cells were located in the mesenchymal tissue in and around the cochlea, and the activated forms were found in and around the spiral ganglion tissue at 37 °C. Our findings suggest that mild hypothermic treatment has a beneficial effect on immune system activation after electrode insertion trauma.

## 1. Introduction

With approximately 2.5 billion people affected by hearing loss and 700 million in need of rehabilitation services, hearing impairment is currently a major global health burden and will continue to be over the coming decades. Cochlear implantation (CI) could be an option for some of these patients. The development of new electrodes in all types of CIs has made remarkable technical progress in the past 30 years [1,2] and is efficacious in children [3] and adults [4]. Despite this progress, electrode insertion is still known to be traumatic for the inner ear, resulting in fibrosis, interstitial proliferation and autoimmune reactions [5,6,7,8,9,10].

Numerous studies have focused on the tissue trauma caused by the insertion of the electrode—also referred to as electrode insertion trauma (IT). On the one hand, IT is directly caused by the insertion, and on the other hand, IT also causes various immune reactions in different tissues. These changes are not well understood. Previously, the inner ear was considered an immune privileged region. A recent study demonstrated the presence of immune cells—especially macrophages—in the human fetal inner ear, emphasizing their important role in the development of the cochlea and vestibulum [11]. In addition, the induction of an inflammatory response in a murine noise-induced acoustic trauma model has previously been described. This activity was explained by the recruitment of circulating immune cells and chemokine and cytokine regulation [5]. In addition, macrophages have been detected in the cochlea without a history of local trauma and/or hearing disorders [12].

IBA1 [13], F4/80 [14], CD163 [13] and CD45 [15] are markers for the presence of macrophages and monocytes in the murine inner ear. 

Ionized binding adaptor molecule 1 (IBA1) is a protein that binds specifically to macrophage and microglia surfaces [13,16]. IBA1 is expressed in the nucleus, small multicellular complexes with anchors to the extracellular matrix, podosomes and cytoplasm [13,17]. IBA1 is involved in the membrane ruffling of macrophages in the phagocytosis and motility processes [13,16,17].

F4/80 is a member of the epidermal growth factor (EGF)-transmembrane 7 (TM7) family. The molecule is expressed on several macrophage subsets and is specific for mouse macrophages. The detailed function of F4/80 has not been fully elucidated. To the best of our knowledge, this molecule is involved in peripheral tolerance to ovalbumin and the inhibition of immune cell responses to peripheral antigens [14,18,19,20]. 

The scavenger receptor molecule CD163 is a specific marker for monocytic cells, such as macrophages, monocytes and microglia. CD163 is present as a transmembrane protein and acts as an endocytic receptor for hemoglobin–haptoglobin complexes [13,21,22]. Glucocorticoids and interleukins can induce CD163 through anti-inflammatory mediators. Furthermore, CD163 is involved in cytokine assembly against bacterial infections and is known as a sensor for bacteria [22,23,24]. 

The leukocyte common antigen marker CD45 is a highly conserved tyrosine phosphatase receptor that can be found on leukocytes and hematopoietic cells such as macrophages and dendritic cells. CD45 is involved in the modulation and regulation of immune cells, as well as the regulation of the development of lymphocytes and macrophages [15,25,26,27]. 

These cell-specific markers represent the variability and range of immune cells involved in CI electrode insertion trauma. IBA1 is a surrogate marker for specific microglia cells. A major aera of interest in this study is the evaluation of the immune response in the neuronal structures of the cochlea, specifically the spiral ganglion. The study was performed using a murine whole-organ cochlea explant culture, which led to the selection of the F4/80 marker. The F4/80 marker is known to react specifically with murine macrophages. A further objective of this study was the evaluation of the CD163 marker. CD163 is a selective monocyte marker and can be induced by anti-inflammatory stimuli, such as glucocorticosteroids. A similar effect can be expected with the application of hypothermia, another postulated anti-inflammatory stimulus [28,29,30]. With the selection of the CD45 marker, the analysis has been extended to the dendritic cells, another important line of immune cells. The CD45 marker is specific to dendritic cells. 

Therapeutic hypothermia has been described as a possible otoprotective method. It can reduce the intracellular inflammatory response after CI implantation. The effect of otoprotection has been correlated to hair cell and nerve fiber preservation [31,32,33,34]. In an experimental study by Tamames et al., the positive effect of therapeutic hypothermia on hearing preservation in electrode insertion trauma was shown [32]. In an in vivo rodent cochlea implantation model, hypothermia preserved the residual hearing in contrast to the normothermic group. Furthermore, the usability of local hypothermia in humans has been investigated by Bader et al., who applied room-temperature rinsing in situ during surgery [31]. This publication suggested an easy and ethically acceptable hypothermia application for the inner ear. With the usability of hypothermia in real clinical otologic settings being proven, further information is needed with respect to the underlying pathophysiologic mechanisms associated with electrode insertion trauma. This topic has been elaborated in detail by Schmutzhard et al. [30]. The authors utilized a whole-organ cochlea culture model to evaluate the effect of mild hypothermia on electrode insertion trauma. Electrode insertion trauma consists of an apoptotic and a fibrotic sequence [35]. In the hypothermia study, these inflammatory pathways could be reproduced in the normothermic group. Interestingly, in hypothermic conditions, the proapoptotic markers were found to be decreased with the application of hypothermia. Accordingly, the antiapoptotic markers were significantly increased in the hypothermic group. This suggested an antiapoptotic effect of hypothermia on the inner ear. Real-time PCR for Tumor Necrosis Factor alpha and Interleukin-I-beta supported this conclusion. However, the effect of the hypothermia effect on the macrophage system is a missing link. 

The aim of the present investigation was to determine the correlation and distribution of activated and non-activated immune cells in the inner ear in an IT model, utilizing a whole-organ cochlea culture technique. The effect of therapeutic hypothermia on the immune response was evaluated, comparing the effects at normothermic (37 °C) and hypothermic (32 °C) conditions. Therapeutic hypothermia is expected to reduce the immune reaction during CI implantation, potentially resulting in an otoprotective effect.

## 2. Results

### 2.1. IBA1 Distribution and Activation Stage

Twelve cochleae were used for IBA1 staining—four untreated controls, four with IT cultured at 37 °C and four with IT cultured at 32 °C. At 32 °C, immune cells were present in the mesenchymal tissue of the cochlea (Figure 1A). In the SPG, 73% of the immune cells (Figure 1A), 67% in the BM/OC (Figure 1A) and 71% in the SL (Figure 1A) were found to be non-activated. In contrast to those cultured at 32 °C, at 37 °C, an accumulation of IBA1-positive cells was observed around and in the modiolus of the cochlea (Figure 1B, arrow). Additionally, the distribution of immune cells shifted from rhizoid (non-activated) to more amoeboid (activated) forms ranging between 60% and 68% depending on the location.

### 2.2. F4/80 Expression and Activation Stage

Twelve cochleae were used for F4/80 staining—four untreated controls, four with IT cultured at 37 °C and four with IT cultured at 32 °C. F4/80 immune labeling of cochleae cultured in hypothermic experimental conditions showed an accumulation of immune positive cells near the modiolus and some spots at the stria vascularis (Figure 2A, arrow). The majority (66–80%) were non-activated forms. At 37 °C, with insertion trauma, the activated forms dominated in proportions ranging from 71 to 73% depending on the location (Figure 2B). 

The untreated control group showed a dominance of non-activated F4/80-positive cells (95–96%) (Figure 2C).

When exposed to therapeutic hypothermia, the macrophages and monocytes showed dendritic filopodia (non-activated) (Figure 2D). In the 37 °C culture condition, a shift of the majority of the cells to the active form indicates a possible activation of phagocytosis (Figure 2D,E).

### 2.3. CD45 Expression in the Mouse Inner Ear

Twelve cochleae were used for CD45 staining—four untreated controls, four with IT cultured at 37 °C and four with IT cultured at 32 °C. Macrophages labeled with the CD45 antibody were detectable in the mesenchymal tissue of the entire cochlea (Figure 3A) at 32 °C. Non-activated forms (between 70% and 73%) were evident in all examined regions—the spiral ganglion, the sensory organ and the spiral ligament (Figure 3A). 

At 37 °C, an accumulation of CD45-positive cells in the cochlea was found in the mesenchymal tissue (Figure 3B). A shift towards more activated forms could be seen in the normothermic group with 67% in the spiral ganglion, 75% in the sensory organ and 71% in the spiral ligament.

In the control group, no particular accumulation in the whole cochlea was evident (Figure 3C). Non-activated cells, with a proportion between 78% and 84%, were found to be the dominating form. 

In hypothermic conditions, the macrophages had filopodia when they were in a non-activated form (Figure 3D, arrow), whereas the activated form showed ameboid shapes (Figure 3E).

### 2.4. CD163 Expression Distribution Is Influenced by Mild Hypothermia

Twelve cochleae were used for CD163 staining—four untreated controls, four with IT cultured at 37 °C and four with IT cultured at 32 °C. CD163-positive macrophages/monocytes at 32 °C were observed in the mesenchymal tissue of the whole cochlea (Figure 4A). The immune cells exhibited more non-active forms (between 67% and 78%) in the spiral ganglion, sensory organ and spiral ligament (Figure 4A).

At 37 °C, the macrophages/monocytes also showed a region-specific accumulation in the modiolus of the cochlea and in the mesenchymal tissue (Figure 4B). Likewise, an increase in more activated forms up to 74% in the spiral ganglion and 60% in the sensory organ basal membrane and organ of Corti was shown for CD163-stained immune cells. 

In the control group, the distribution of CD163-positive cells in the mesenchymal tissue of the cochlea was observed (Figure 4C). Both macrophages and monocytes showed a reduced percentage of non-activated shapes, between 79% and 86%, similar to the results of the CD45 staining. 

The immune cells labeled with CD163 and cultured at 32 °C had a polygonal and/or triangular shape (Figure 4D, arrow) and at 37 °C, they had a rounded appearance (activated) (Figure 4E).

## 3. Discussion

The presented immunohistochemical examination identified macrophages and monocytes after short-term cultivation at 32 °C and 37 °C. The total number of macrophages was comparable in both study groups as well as the control group. Interestingly, the number of activated and non-activated forms differed significantly. The insertion trauma (IT) at 37 °C resulted in a majority of macrophages being activated. In contrast, the 32 °C group showed a predominance of the non-activated forms. A similar expression pattern was observed in the untreated control group. This implicates an activation stimulus of IT under normothermic conditions. Hypothermia seems to have the capacity to antagonize this effect. 

At 32 °C, immune cells were present in the whole cochlea, without a visible accumulation in a specific region in the inner ear for IBA1, CD163 or CD45. In the fluorescence staining of F4/80 at 32 °C and 37 °C, the macrophages and monocytes could be detected everywhere in the cochlear tissue, but with a higher accumulation in and around the nerve tissue of the cochlea, as seen in the quantification of stained macrophages/monocytes independent of their activated and non-activated form (Figure 5). 

The basal membrane, organ of Corti (Figure 6), spiral ligament and stria vascularis (Figure 7) presented a lower number of stained immune cells than the spiral ganglion. The accumulation of macrophages and monocytes at 32 °C (only for F4/80) and at 37 °C can be explained by the fact that macrophages appear to protect the nerve tissue after a nerve injury [36]. In several rodent studies, neural macrophages were shown to be important factors in neuroinflammation and for the progression of neurodegeneration. The interaction between neurodegeneration and immune response in neuronal disease, degeneration, development and homeostasis response will be an important part of future research [37]. Macrophages at 37 °C and F4/80-positive macrophages at 32 °C accumulated close to the spiral ganglion because the nerves at the habenula perforata lost their compact myelin [38]. Macrophage accumulation indicates degeneration of nervous tissue during the process of insertion. Hence, macrophages remove the cell debris from damaged nerve fibers.

Cochlea implant electrodes have been improved significantly in the past few decades with the aim of reducing insertion trauma. Hence, hearing and structure preservation results have improved significantly in the past few years. Despite these improvements, significant loss of residual hearing can still occur immediately after surgery or in the following months. The cause of this is not fully understood, though it is thought to be triggered independently from the chosen electrode and surgical technique [39]. This finding supports the necessity of furthering our understanding of the pathophysiology of IT. Strategies to reduce the inflammatory response after cochlear implant surgery and preserve the residual hearing ability use dexamethasone-coated electrodes [40] or otoprotection via mild hypothermia [31]. Dexamethasone-eluting electrodes protect residual hearing post cochlear implant surgery [41]. CI implantation results in an increased inflammatory response around the electrodes [40].

One of the postulated main functions of therapeutic or prophylactic hypothermia is the reduction in the release of apoptotic and proinflammatory molecules, which leads to destruction of the tissue [42,43,44]. Hypothermia is able to protect against noise-induced hearing loss in mice [45,46]. Additionally, a reduction in glutamate-induced excitotoxicity has been identified [47]. In guinea pig cochleae, hypothermia showed a positive effect on the reduction in potassium ion concentrations, which has an impact on the blood–labyrinth barrier [48]. Hypothermia could inhibit a disruption to similar barriers and decrease their vascular permeability [49,50]. In the present study, the reaction of the immune system caused by implant surgery was studied with regard to a possible otoprotective effect of therapeutic hypothermia. At 32 °C, the immune cells were more densely accumulated in the mesenchymal tissue in and around the cochlea. At 37 °C, the immune cells accumulated in and around the spiral ganglion tissue. The positive effect of mild hypothermia resulted in a higher percentage of non-activated immune cells in the cochlea. The production of cytokines and inflammatory signals and the recruitment of other immune cells or the reduction in apoptotic pathways could explain this observation [51]. Hypothermia can decrease neutrophil phagocytosis, motility and production of reactive oxygen. Mild hypothermia has a positive effect on the expression of cytokines that are released by peripheral blood mononuclear cells and affects the release of the inflammatory cytokines IL-2, IL-10 and IFN [52,53]. Reduced production of inflammatory factors can lead to better preservation of residual hearing.

The ionized binding adaptor molecule 1 (IBA1) staining showed an accumulation of macrophages and monocytes at 37 °C in/around and near the spiral ganglion. Our results suggest that the nerve cells suffer at 37 °C from implantation stress, leading to faster degeneration where the removal of cellular debris is important. During apoptosis, the cells release several cytokines to recruit active microglia/macrophages in order to remove the dead cells [13,16,54]. This explains the presence of more “active” immune cells at 37 °C. Meanwhile, with therapeutic hypothermia, apoptotic stress is reduced as well as the necessity of releasing “active” immune cells. The mouse macrophage F4/80 immune staining showed the opposite: temperature-dependent accumulation. More cells were observed around/near the spiral ganglion with mild hypothermia. F4/80 is involved in the process of wound healing, inducing the release of chemoattractants/chemokines to recruit other immune cells [14,20,55]. Further, it is involved in cell–cell interactions, which are important for the healing process [19]. Immune cell accumulation with therapeutic hypothermia depends on the release of more inflammatory factors to induce the wound-healing process [20,23,55]. The leukocyte surface marker CD45 and scavenger receptor molecule CD163 showed no (CD45) or only low-density (CD163) accumulation near/around the spiral ganglion for both temperatures [13,15]. The distribution of activated and non-activated macrophages after therapeutic hypothermia showed more non-activated cells. The higher percentage of non-activated cells can be explained by the fact that for CD45, at 32°C, the inflammatory response is reduced or inhibited, thereby leading to a reduced recruitment of CD45-positive leukocytes [56], which are essential for the regulation of the signal transduction pathway of immune cells during an immune reaction [26]. A reduction or inhibition of pro-inflammatory molecules, such as glucocorticoids and IL-10, is a strong stimulus for CD163-expressing immune cells. This was observed for the stained cells in the therapeutic hypothermia conditions [13,24]. 

This study described an overview of the immune reaction and distribution of activated and non-activated immune cells in a whole-cochlea culture insertion trauma model. The reduction in activated macrophages in the hypothermia group sheds some light on the pathogenic mechanism involved in the otoprotective effect of therapeutic hypothermia. 

## 4. Materials and Methods

### 4.1. Subjects

*C57BL/6J mice* (both male and female, the Jackson Laboratory, Bar Harbor, ME, USA) were used in this investigation. Breeding mice were housed at the Innsbruck’s Medical University Animal Facility and were protected from noise exposure during the investigation. Deeply anesthetized animals were euthanized by means of rapid cervical dislocation. The extraction of tissue from the euthanized animals conformed with the Austrian Federal Acton Experiments of Living Animals (Tierversuchsgesetz 2012–TVG 2012, §2) based on the EU Directive 2010/63/EU and is not regarded as an animal experiment. No human studies are presented in this manuscript. No potentially identifiable human images or data are presented in this manuscript.

### 4.2. Tissue Collection and Preparation

Three different study groups were considered. The first category included untreated cochleae with no culturing and no IT and was used as a control group to visualize the immune cell status without IT. In the two other groups, IT was inflicted prior to culturing. Then, these two groups were cultured at 37 °C and 32 °C. A total of 48 cochleae (12 mice per antibody with 4 per study group) were extracted from 10-day-old mice within 3 min and divided into the three study groups. Afterwards, a fishing line was inserted into the previously opened round window through the basal turn of the cochlea, imitating electrode insertion trauma in the two IT groups. 

Then, the cochlea samples were divided into the following study groups: whole-organ cochlea culture for 24 h at 37 °C and 32 °C with insertion trauma and an uncultured control group without insertion trauma. The two culturing groups were treated for 24 h and fixed according to the protocol described in Appendix A. Afterwards, the cochleae were decalcified in 20%EDTA and cryo-embedded according to standard procedures [57]. The cochlea tissues were sectioned into 5 µm sections using a Leica CM 3050 microtome (Leica Biosystems, Richmond, IL, USA).

### 4.3. Immune Labeling

The fluorescence staining of mouse inner ear sections was performed using a Ventana Roche^®^ Discovery XT Immunostainer (Mannheim, Germany) after a standard fluorescence staining procedure for cryosections from Ventana. The program was pre-programmed with one-hour incubations of the primary and secondary antibodies. After completion of the staining run, DAPI staining, phalloidin–FITC staining and quenching (Vector^®^ TrueView^®^ Autofluorescence Quenching Kit, Vector Laboratories^®^, Burlingame, CA, USA) were performed according to the manufacturer’s instructions. The sections were mounted using Vectashield Antifade Mounting Medium (Vector Laboratories^®^, Burlingame, CA, USA). The hosts, dilutions and sources of the primary and secondary antibodies used are listed in Appendix A.

### 4.4. Image Acquisition and Quantitative Data Analyses

The quantitative analyses were performed on a total of 48 cochlea with the following distribution: 12 per antibody with 4 for each study group. Furthermore, 5 to 10 representative midmodiolar sections from each cochlea were evaluated. The fluorescence-stained sections were digitalized at 40× magnifications using TissueFAXS Plus microscope software (https://www.google.at/url?sa=t&rct=j&q=&esrc=s&source=web&cd=&cad=rja&uact=8&ved=2ahUKEwiFp5jo_Pj-AhXz8bsIHZsoBBAQFnoECBUQAQ&url=https%3A%2F%2Ftissuegnostics.com%2Fproducts%2Ffluorescence-brightfield-cytometer%2Ftissuefaxs-plus&usg=AOvVaw0o7RR4WlV_9h394fJZLvUL, accessed on 22 March 2023) coupled to a Zeiss^®^ Axio Imager Z2 microscope (Jena, Germany). Selected slides were imaged using a Leica SP8 STED confocal microscope (Leica Microsystems, Wetzlar, Germany). Two different morphological types of monocytes/macrophages were distinguished: non-activated (N.-A.) and activated (A.). Non-activated macrophages have a triangular or rhizoid appearance and many lopodia. Activated macrophages can be directly discerned by their round spherical shape and whether they are observed phagocytosing other apoptotic cells. The cell count was performed separately on three different anatomical compartments: the spiral ganglion (SPG), the spiral ligament and the stria vascularis. Two independent specialists rechecked and confirmed the cell counts performed in TissueQuest (Version 7.0.1.148 by Tissuegnostics, Vienna, Austria). For the manual cell counts, TissueQuest (Version 7.0.1.148 by Tissuegnostics, Vienna, Austria) was used. The absolute cell counts in the different tissues were summarized and calculated as a percentage of activated or non-activated values in Graphpad Prism (Version 9.4.1 (681) by GraphPad Software©, San Diego, CA, USA). 

## Figures and Tables

**Figure 1 ijms-24-08850-f001:**
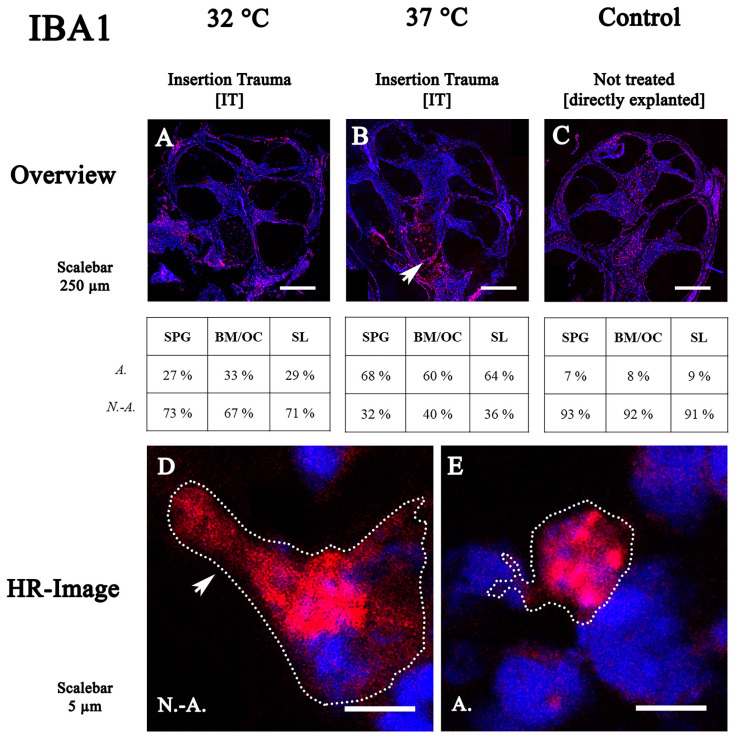
Fluorescence staining of IBA1 in cochleae cultured at 32 °C and 37 °C. (**A**–**C**): overview of midmodiolar mouse inner ear sections at 32 °C or 37 °C with insertion trauma and control group. (**A**) at 32 °C, the macrophages and monocytes were observed in the mesenchymal tissue of the cochlea. (**B**) at 37 °C, an infiltration into the spiral ganglion tissue could be observed (arrow). (**C**) in the control group, no specific accumulation was seen. (**D**,**E**) high-resolution confocal images of spiral ligament and stria vascularis. Macrophages and monocytes were present in the spiral ligament and connective tissue of the stria vascularis. Non-activated cells are triangular or polygonal in shape and have certain dendritic filopodia. The presence of activated macrophages indicates phagocytosis of cells. SPG: spiral ganglion; BM: basal membrane; OC: organ of Corti; SL: spiral ligament; N.-A.: non-activated; A.: activated.

**Figure 2 ijms-24-08850-f002:**
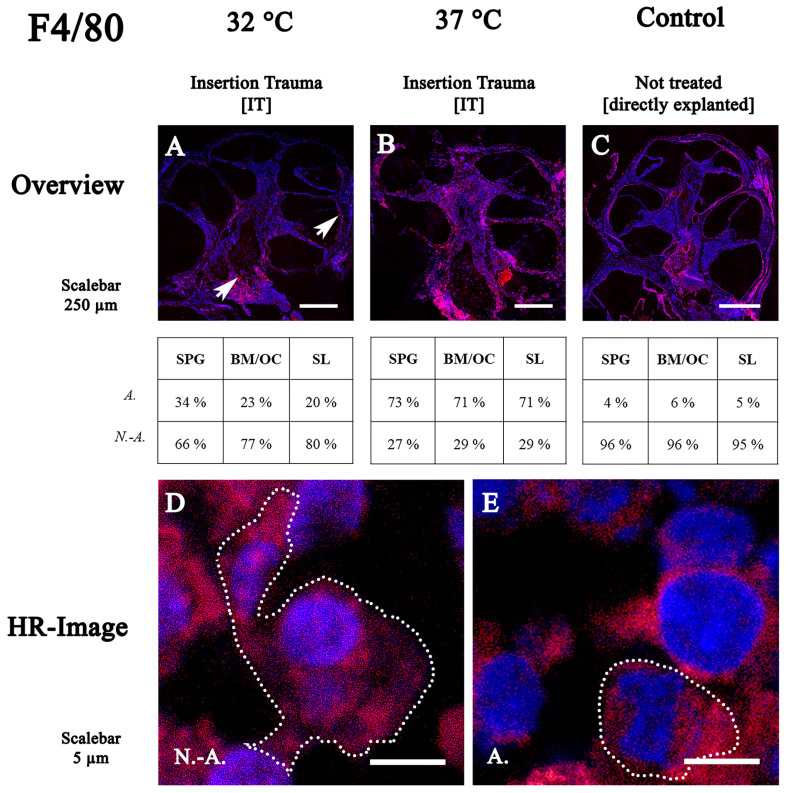
Fluorescence staining of F4/80 in cochleae cultured at 32 °C and 37 °C. (**A**–**C**) overview of midmodiolar mouse inner ear sections at 32 °C or 37 °C with insertion trauma and untreated control group (**D**,**E**) high-resolution confocal images of spiral ligament and stria vascularis macrophages/monocytes. Non-activated cells are polygonal in shape. Activated macrophages are round in shape and can be observed phagocytosing a cell. SPG: spiral ganglion; BM: basal membrane; OC: organ of Corti; SL: spiral ligament; N.-A.: non-activated; A.: activated.

**Figure 3 ijms-24-08850-f003:**
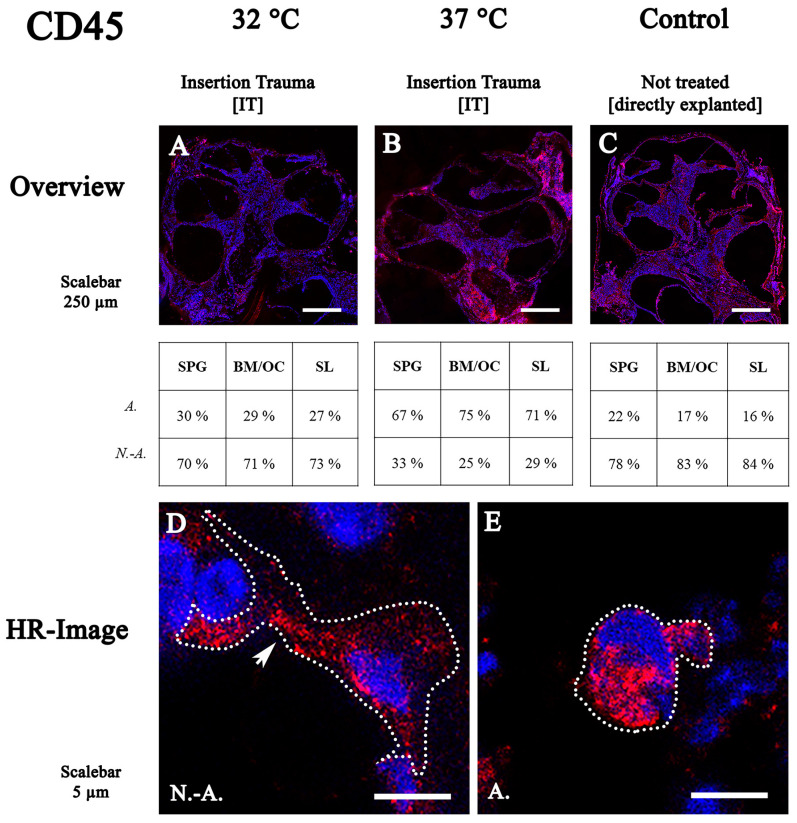
Fluorescence staining using CD45 antibody of cochleae cultured at 32 °C and 37 °C. (**A**–**C**) midmodiolar mouse inner ear sections with insertion trauma at 32 °C/37 °C and untreated control group. (**D**,**E**) high-resolution confocal images of spiral ligament and stria vascularis. Macrophages and monocytes are present in the spiral ligament and connective tissue of the stria vascularis. Non-activated cells are triangular or polygonal in shape. Activated macrophages are round in shape and can be observed phagocytosing a cell. SPG: spiral ganglion; BM: basal membrane; OC: organ of Corti; SL: spiral ligament; N.-A.: non-activated; A.: activated.

**Figure 4 ijms-24-08850-f004:**
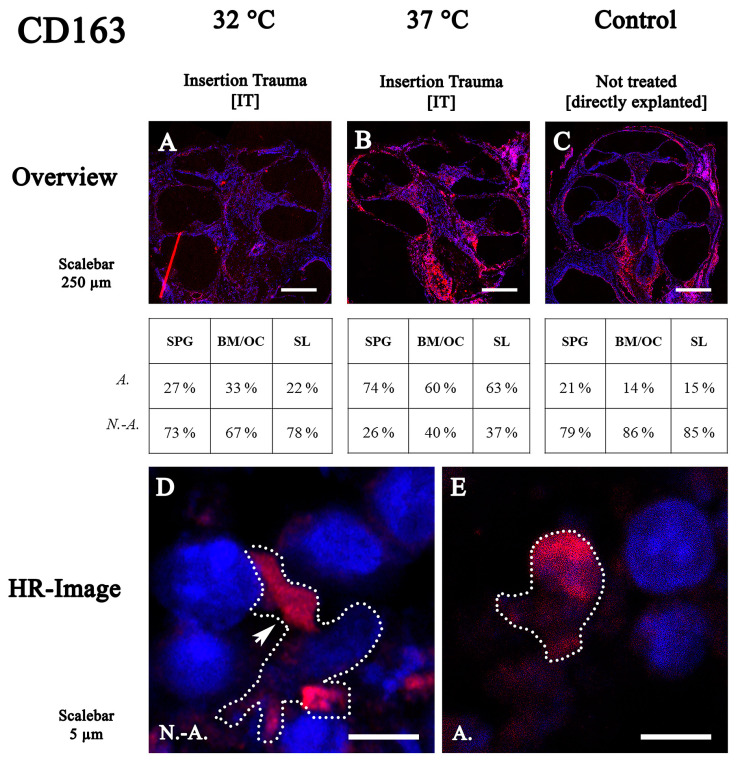
Fluorescence staining of CD163 in cochleae cultured at 32 °C and 37 °C. (**A**–**C**) midmodiolar mouse inner ear sections with insertion trauma at 32 °C or 37 °C and untreated control group. (**D**,**E**) high-resolution confocal images of spiral ligament and stria vascularis. Non-activated cells are triangular or polygonal in shape and have short filopodia. Activated macrophages have round shapes. SPG: spiral ganglion; BM: basal membrane; OC: organ of Corti; SL: spiral ligament; N.-A.: non-activated; A.: activated.

**Figure 5 ijms-24-08850-f005:**
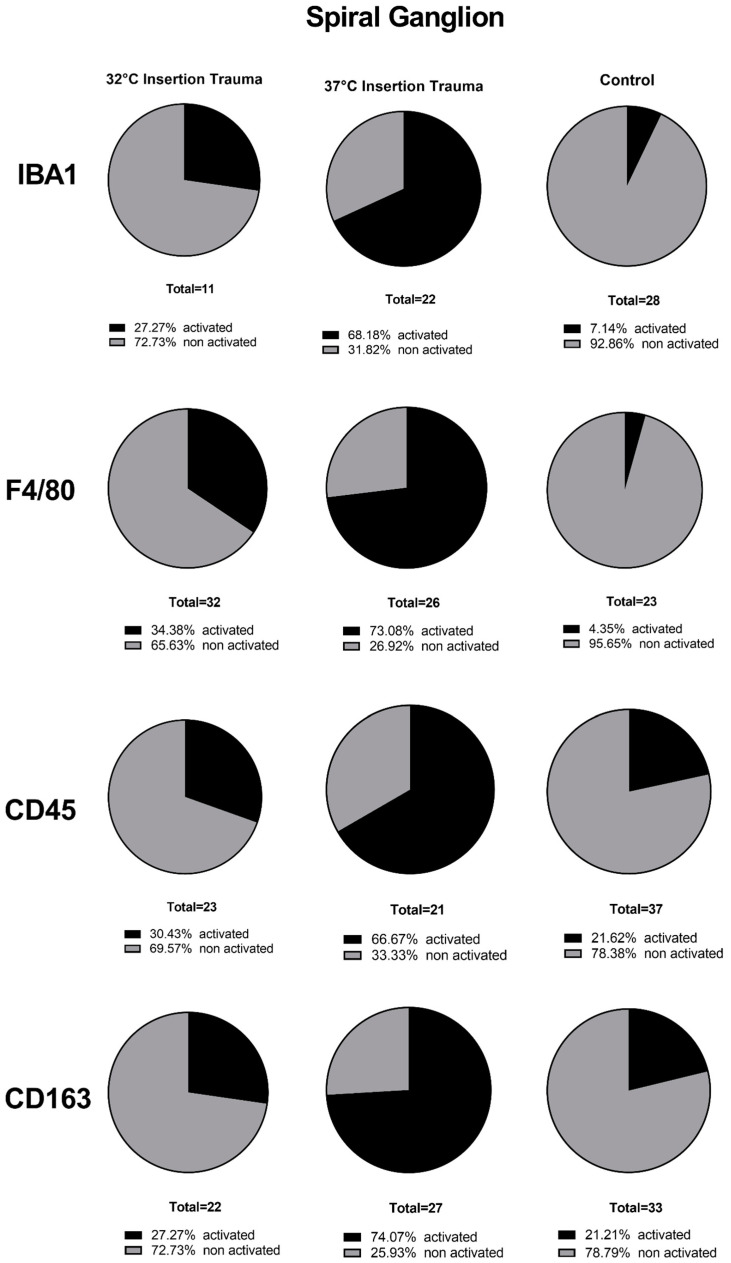
Quantification of macrophages/monocytes in spiral ganglion of 10-day-old mice. For IBA1, at 32 °C, there were more non-activated than activated immune cells; at 37 °C, more activated forms than non-activated forms were present. The highest percentage of non-activated immune cells was displayed by the control group. For F4/80, CD45 and CD163, and between groups, there were similar distribution trends of activated and non-activated immune cells, as described for IBA1.

**Figure 6 ijms-24-08850-f006:**
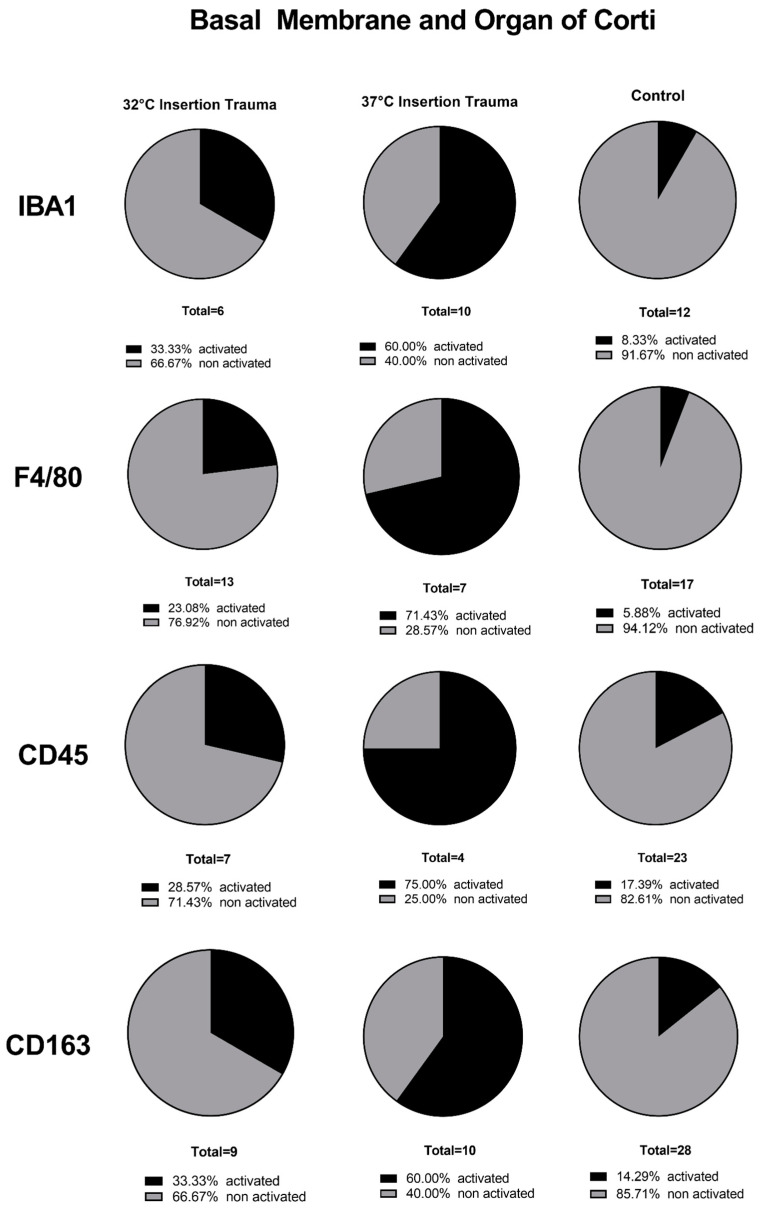
Quantification of macrophages/monocytes in basal membrane and organ of Corti of 10-day-old mice. The effect of mild hypothermia (comparing 32 °C versus 37 °C versus control group) displayed the expected distribution of more non-activated immune cells at the lower temperature and more activated immune cells at body temperature. These proportional distributions were seen with all four antibodies (IBA1, F4/80, CD45 and CD163).

**Figure 7 ijms-24-08850-f007:**
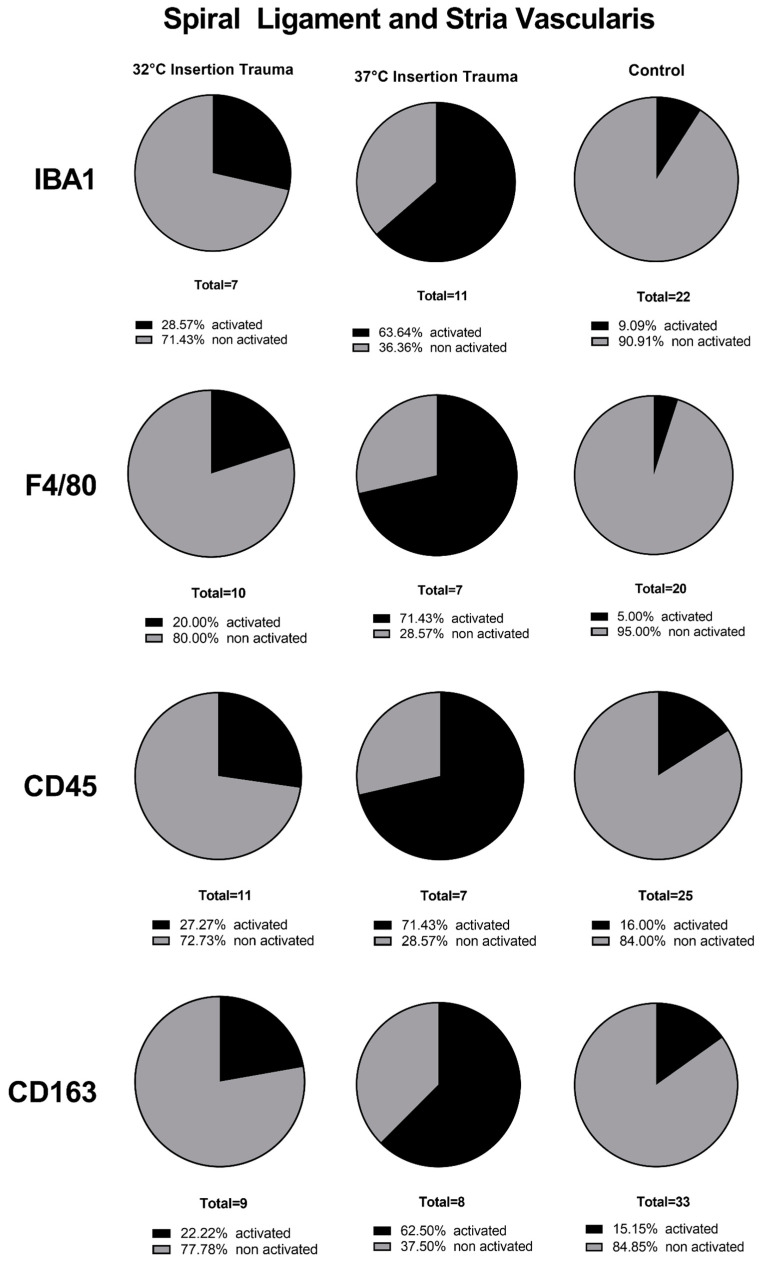
Quantification of macrophages/monocytes in spiral ligament and stria vascularis of 10-day-old mice. The effect of mild hypothermia displayed the expected distribution of more non-activated immune cells at 32 °C and more activated immune cells at 37 °C. These proportional distributions were seen with all four antibodies (IBA1, F4/80, CD45 and CD163).

## Data Availability

The data presented in this study are available on request from the corresponding author. The data are not publicly available due to institutional guidelines.

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
