# Peer review of "Effects of Therapeutic Hypothermia on Macrophages in Mouse Cochlea Explants"

_ijms, 2023, doi:10.3390/ijms24108850_

Round 1

Reviewer 1 Report

The authors of the manuscript, “Effects of therapeutic hypothermia on macrophages in mouse cochlea explants” have tried to investigate the immunological response after cochlear implants especially macrophages and whether hypothermia is beneficial after electrode insertion trauma in mice” This is a good piece of research but needs clarification/revision on some of the following points.

1. The rationale of selecting these four transmembrane adaptor, marker, protein and receptor is missing in the text which is an important issue for such a study, otherwise, immune response has been analysed comprehensively by Sangaletti et al. (2023) and some other researchers (PMIDs: 36586170, 32317718). What prompted authors to select these four candidates as immune responders? Authors must write it clearly in the introduction section.

2.   The aim of this research is not clear. Authors should be very specific while writing it. The aims written in abstract and in introduction are not similar. So rewrite it.

3.   Reference citation style must be checked in the whole text, as I feel that, instead of citing references as 5;6;7;8;9;10 (line 34), it should be written as [5-10].

4.   There are many grammatical mistakes in the text, so it must be corrected by English language expert.  In the line 16 of abstract “will be evaluated” should be “was evaluated”, Line 39- “present” should be “presence”, line 185-“independed’ should be “independent” so on and so forth.

5.   Full forms of abbreviations should be written in the text, where these come for the first time.

Author Response

To whom it may concern

Thank you very much for the opportunity to comment on the reviewers´ statements.

The following changes and comments have been made according to the assessment of Reviewer 2 and are marked in red in the manuscript:

  1. The rationale of selecting these four transmembrane adaptor, marker, protein and receptor is missing in the text which is an important issue for such a study, otherwise, immune response has been analysed comprehensively by Sangaletti et al. (2023) and some other researchers (PMIDs: 36586170, 32317718). What prompted authors to select these four candidates as immune responders? Authors must write it clearly in the introduction section.

The motivation for the selected immune markers has been claridfied in the introduction section. Line 69 to 80

  1. The aim of this research is not clear. Authors should be very specific while writing it. The aims written in abstract and in introduction are not similar. So rewrite it.

The aims of the study have by synchronized in the abstract

Line 15-19

  1. Reference citation style must be checked in the whole text, as I feel that, instead of citing references as 5;6;7;8;9;10 (line 34), it should be written as [5-10].

The Citation style has been adapted

  1. There are many grammatical mistakes in the text, so it must be corrected by English language expert.  In the line 16 of abstract “will be evaluated” should be “was evaluated”, Line 39- “present” should be “presence”, line 185-“independed’ should be “independent” so on and so forth.

The English has been proofread by an medical writer.

  1. Full forms of abbreviations should be written in the text, where these come for the first time.

Has been performed.

Reviewer 2 Report

The current original study aimed to find out the correlation and distribution of activated and non-activated immune cells in the inner ear in an insertion trauma mouse model, utilizing whole organ cochlea culture technique. The effect of therapeutic hypothermia was evaluated on the immune response, comparing the effects of normothermic and hypothermic conditions. The Introduction, Results and Discussion part are well-written and conclusive, with minor mistypes to be corrected. The Material and methods part is detailed enough to understand. Figure legends are also accurate and easy to follow.

I suggest English language quality improvement, maybe you could give it to a native speaker for the revision of language errors.

Author Response

The following changes and comments have been made according to the assessment of Reviewer 2 and are marked in blue in the manuscript:

If the reviewers or editor recommended English language editing, this can be arranged by MDPI. Note that language editing by MDPI is not compulsory, nor does it guarantee that your manuscript will eventually be accepted for publication. Click on the link for more information and to request a quotation.

  1. I suggest English language quality improvement, maybe you could give it to a native speaker for the revision of language errors.

Thank you for the kind review. The English has been proofread by an medical writer.

Reviewer 3 Report

The study is interesting, but very poorly presented. Some parts are hard to understand. There are misspelled words, missing punctuation... it almost feels like the manuscipt is a draft that hasn't been re-read.

The groups (i.e. how many mice for the different groups) must be described in the materials and methods, in order to understand the statistical significance of the study and the data presented. From this it would be possible to understand whether the data obtained is random or not. And whether immune activation can, in fact, be controlled with an action of hypothermia.

The perplexity arises from the fact that most cochlear implants are made in such a way as to avoid insertion trauma. So, is there really any clinical utility?

However, the part of the results needs to be rewritten and improved because only through the supplementary data is it possible to truly understand what the situation is at the 2 different temperatures. Instead, this should be inferred from the main text.

Author Response

...

Round 2

Reviewer 3 Report

The revised work is now of excellent quality and very interesting. Congratulations to the authors.

Author Response

Thank you very much for your positive remark.